# The Effects of Time Restricted Feeding on Overweight, Older Adults: A Pilot Study

**DOI:** 10.3390/nu11071500

**Published:** 2019-06-30

**Authors:** Stephen D. Anton, Stephanie A. Lee, William T. Donahoo, Christian McLaren, Todd Manini, Christiaan Leeuwenburgh, Marco Pahor

**Affiliations:** 1Department of Aging and Geriatric Research, Institute on Aging, University of Florida, Gainesville, FL 32611, USA; 2Department of Clinical and Health Psychology, University of Florida, Gainesville, FL 32611, USA; 3Department of Medicine, Diabetes and Metabolism, University of Florida College of Medicine, Gainesville, FL 32611, USA

**Keywords:** weight loss, intermittent fasting, fat loss, sarcopenia, body composition

## Abstract

A growing body of evidence indicates that time restricted feeding (TRF), a popular form of intermittent fasting, can activate similar biological pathways as caloric restriction, the only intervention consistently found to extend healthy lifespan in a variety of species. Thus, TRF may have the potential to also improve function in older adults. Given the challenges many individuals have in following calorie restriction regimens over long-time periods, evaluation of alternative approaches that may produce weight loss and improve function in overweight, older adults is important. Ten overweight, sedentary older adults (≥65 years) at risk for, or with mobility impairments, defined by slow gait speed (<1.0 m/s) participated in this trial. All participants received the intervention and were instructed to fast for approximately 16 h per day over the entire four-week intervention. Outcomes included changes in body weight, waist circumference, cognitive and physical function, health-related quality of life, and adverse events. Adherence levels were high (mean = 84%) based on days goal was met, and mean weight loss was 2.6 kg (*p* < 0.01). Since body composition was not measured in this study, it is unclear if the observed weight loss was due to loss of fat mass, muscle mass, or the combination of fat and muscle mass. There were no significant changes in other outcomes; however, there were clinically meaningful changes in walking speed and improvements in quality of life, with few reported adverse events. The findings of this pilot study suggest that time restricted feeding is an acceptable and feasible eating pattern for overweight, sedentary older adults to follow.

## 1. Introduction

Aging is associated with a host of biological changes that contribute to a progressive decline in cognitive and physical function, ultimately leading to a loss of independence and increased risk of mortality. These changes appear to be accelerated by excess body weight, as well as a lifestyle characterized by low levels of physical activity and/or excessive calorie intake, placing overweight, older adults at increased risk of functional decline [1] For these reasons, there is great interest in the potential role that lifestyle changes may have in slowing rate of functional decline.

To date, caloric restriction (i.e., a reduction in caloric intake without malnutrition) is the only non-genetic intervention that has consistently been found to extend both mean and maximal life span across a variety of non-human species [2]. In overweight humans, caloric restriction has been shown to reduce several cardiac risk factors [3,4], improve insulin-sensitivity [5], and enhance mitochondrial function [6]. Additionally, prolonged caloric restriction has also been found to reduce oxidative damage to both DNA and RNA [7,8,9]. Despite these health-promoting biological changes, most individuals have difficulty engaging in caloric restriction over the long-term and regain weight that was previously lost [10]. For this reason, there is a need to explore the potential of alternative approaches for reducing body weight, specifically body fat, in overweight, older adults at risk for functional decline.

One alternative dietary approach that may produce similar biological changes as caloric restriction that has received increasing interest from the scientific community is intermittent fasting [11]. In contrast to traditional caloric restriction paradigms, food is not consumed during designated fasting time periods but is typically not restricted during designated eating time periods. The length of the fasting time period can also vary but is frequently 12 or more continuous hours. Evidence that intermittent fasting approaches may have beneficial effects on health and longevity first appeared several decades ago [12]. Since this time, a growing body of literature suggests that fasting periods and intermittent fasting regimens can trigger similar biological pathways as caloric restriction (e.g., increased autophagy and mitochondrial respiratory efficiency), which can result in a host of beneficial biological effects including increased circulation and cardiovascular disease protection, modulation of reactive oxygen species and inflammatory cytokines, as well as antimutagenic, antibacterial, and anticarcinogenic effects [13]. 

Despite the known benefits of intermittent fasting within the scientific community, there has recently been a resurgence of interest in intermittent fasting among the lay public. Many claims are being made by health enthusiasts about the potential of intermittent fasting to improve health, and as a result, a growing number of individuals are experimenting with different types of intermittent fasting approaches for their purported benefits. Thus, there is a need to carefully assess the risks and benefits of particular types of intermittent fasting approaches to determine which approaches may or may not be appropriate for specific populations. This is especially true and important when considering the potential risks and benefits of different types of intermittent fasting approaches in older adults and other vulnerable populations.

To our knowledge; however, no study to date has examined the effects of a time restricted feeding (TRF) intervention in overweight, older adults. Given the known loss of lean mass that occurs during both aging and with continuous calorie restriction, TRF regimens may be an effective and potentially sustainable approach to help overweight, older adults lose unhealthy weight. Thus, the primary aims of this pilot study called “Time to Eat” were to evaluate the safety and feasibility of TRF in an overweight, older adult population.

## 2. Materials and Methods

### 2.1. Participants

Participants were overweight, sedentary older adults (≥65 years) with mild to moderate functional limitations. Eligibility requirements included men and women aged 65 years and older, self-reported difficulty walking ¼ mile or climbing a flight of stairs, self-reported sedentariness (<30 min structured exercise per week), walking speed < 1 m/s on the 4 m walk test, able to walk unassisted (cane allowed), and have a body mass index between 25–40 kg/m^2^ (inclusive). Participants who were unwilling or unable to give informed consent or participating in another research project were not accepted. Potential participants were excluded during screening if their phone interview, medical history, or clinical examination revealed any of the following conditions: fasting > 12 h per day, actively trying or planning to lose weight, weight loss >5 lbs. in the past month, resting heart rate of > 120 beats per minute, systolic blood pressure > 180 mmHg or diastolic blood pressure of > 100 mmHg, unstable angina, heart attack or stroke in the past three months, continuous use of supplemental oxygen to manage a chronic pulmonary condition or heart failure, rheumatoid arthritis, Parkinson’s disease, currently on dialysis, active treatment for cancer in the past year, insulin dependent diabetes mellitus, taking medications that preclude fasting for 16 h, or had any condition that in the opinion of the investigator would impair ability to participate in the trial or pose undue personal risk.

### 2.2. Ethics

The protocol for this study received ethics approval from the University of Florida IRB (reference: 201801293) and has been registered at clinicaltrials.gov under reference number NCT03590847 (date: 15 October 2018). This study was conducted in accordance with the ethical principles expressed in the Declaration of Helsinki.

### 2.3. Study Design and Procedures

The primary aims of this pilot study were to evaluate the safety and feasibility of TRF in overweight, sedentary older adults over a four-week period using a single arm design. This pilot study was implemented to obtain preliminary data to allow for refinement of the design, recruitment yields, target population, adherence, retention, safety, and tolerability for future larger trials.

Participants were recruited from the general population in the North/Central Florida area through general mailings and advertisements and targeted outreach to those who have consented to participate in the Claude D. Pepper Recruitment Registry (IRB#417-2007). Following telephone screening, potentially eligible persons were invited to attend a screening visit during which the purposes and procedures of the study were explained, and informed consent was obtained. After the participant provided consent, the following measurements were taken to determine eligibility: (1) height, weight and girth, (2) resting heart rate and blood pressure, (3) 4 m walk test, and (4) medical history. Eligible participants were then invited to attend a baseline assessment visit where they provided a blood sample and completed the following assessments: (a) body weight, (b) waist circumference, (c) walking speed (assessed by six-min walk test), (d) grip strength, (e) cognitive function (assessed by the MoCA), (f) health related quality of life (HRQoL; assessed by SF-12), and (g) fatigability (assessed by the Pittsburg Fatigability Scale).

To assess feasibility and acceptability of the TRF intervention within this study population, changes in physical and cognitive function, body weight, grip strength, perceived fatigability, self-reported HRQoL, and adverse events were evaluated. All baseline assessments were repeated at the week four follow-up visit, and diaries were collected. Data were collected at the University of Florida’s Institute on Aging Clinical Translational Research Building.

### 2.4. Intervention

Following completion of the baseline assessments, participants received instruction on the ‘Time to Eat’ intervention in which they were instructed to fast for approximately 16 h per day for a period of four weeks with the daily target range between 14–18 h. The first week involved a ramp up to a full 16-h fasting period (Days 1–3 fast for 12–14 h per day, Days 4–6 fast for 14–16 h per day, Days 7–28 fast for 16 h per day). Participants were allowed to consume calorie-free beverages, unsweetened teas, black coffee, sugar-free gum, and were encouraged to drink plenty of water throughout the entire intervention period. Participants were provided a written instruction sheet to inform them of which beverages were ok to consume while fasting and which were not. They were also provided with a food diary and instructed to record the time of their first and final calorie consumption each day. There were no dietary restrictions on the amount or types of food consumed during the 8-h feeding window, and participants were allowed to choose the time frame that best fit their lifestyle.

Participants were contacted via phone by the study interventionist at the end of weeks 1, 2, and 3. The purpose of these calls was to review the protocol, monitor for adverse events, and provide support and guidance to promote adherence to the intervention. At each follow-up contact, participants were asked about any changes to their health or physical function since the previous contact and any changes were documented on the adverse event log. Participants were also asked about problems or challenges in following the study intervention.

### 2.5. Adherence

Adherence to the study intervention was measured using food diaries. Participants were considered compliant to the study intervention if they fasted between 14–18 h per day during weeks 2–4.

### 2.6. Outcomes

#### 2.6.1. Anthropometric and Metabolic Measures

Body Weight. Body weight was measured following the removal of excess clothing and shoes with a calibrated scale (Detecto).

Waist Circumference. Waist circumference was taken at the mid-point between the participant’s lowest rib and the top of his/her hip bone.

Blood Glucose. Blood was drawn from participants in a fasted state. Glucose levels were measured by Quest Diagnostic Clinical Laboratories, which is accredited by the College of American Pathologists.

Blood Pressure. Resting systolic and diastolic blood pressure were taken after participants spent 5 min seated in a quiet room, free of distractions. Blood pressure was obtained according to a standardized protocol [14]. Three readings of blood pressure, spaced one minute apart, were taken using a sphygmomanometer with appropriate cuff size. The first reading was discarded, and the last two readings were averaged. If large differences were observed between the second and third readings, an additional reading was taken and the median value for the three trials was used.

#### 2.6.2. Physical Function Measures

Six Minute Walk. The six-minute walk test measures the amount of distance the participant can complete on a standard walking course in six minutes without running or overexerting themselves. Participants were instructed to walk at a fast pace and the distance covered in meters was measured.

Grip Strength. Participants were asked to squeeze the dynamometer as hard as possible with their dominant hand. Two measurements were taken, and the average was used in analyses. Isometric hand grip strength is a commonly used measure of upper body skeletal muscle function and is widely used as a general indicator of functional status [15].

#### 2.6.3. Cognitive Function Measures

Montreal Cognitive Assessment (MoCA). The MoCA is a 30-point assessment of mild cognitive impairment, which assesses the domains of attention and concentration, executive functions, memory, language, visuoconstructional skills, conceptual thinking, calculations, and orientation [16]. Different versions of the test were given at baseline and week four to avoid learning effects between repeated administrations of the test.

#### 2.6.4. Health Related Quality of Life

Short Form (SF) -12 Health Survey. The SF-12, a 12-item health questionnaire, was used to assess several domains of HRQoL, including physical health status, mental health status, and general health perception [17]. Scores on this measure range from 0 to 100, with greater scores representing better health.

#### 2.6.5. Fatigability

The Pittsburgh Fatigability Scale. This 26-item self-administered questionnaire was used to measure perceived mental and physical fatigability [18].

### 2.7. Statistical Methods

This trial represented a pilot study, which was designed to assess the feasibility, acceptability, and efficacy of a TRF intervention in overweight, sedentary older adults; therefore, a power analysis was not conducted. The statistical analyses consisted of paired sample two-tailed *t*-tests. The means and standard deviations of variables were computed at baseline and at the end of the study; for responses with missing values, the values were not included in the analyses. Change from baseline was defined as the value at time *t* minus the value observed at baseline for all response measures. The outcomes of interest were change from baseline to week four on the following measures: (a) body weight, (b) waist circumference, (c) walking speed (assessed by six-min walk test), (d) grip strength, (e) cognitive function (assessed by the MoCA), (f) HRQoL (assessed by SF-12), and (g) fatigability (assessed by the Pittsburg Fatigability Scale). All analyses were conducted using SPSS (version 25; IBM Corp, New York, NY, USA). 

## 3. Results

Ten overweight, sedentary older men and women (mean age = 77.1 years; 6 women and 4 men) with mild to moderate functional limitations participated in this trial. Participant flow during the trial is outlined in Figure 1. One participant dropped out due to personal health issues unrelated to the intervention.

Self-reported mean adherence to the TRF regimen was 84%, measured by daily time diaries. Participants reported fasting for approximately 15.8 h daily during the intervention period. Mean weight loss was 2.6 kg (*p* < 0.01) with weight loss occurring in eight out of nine participants. There were no significant changes on cognitive or physical function measures; however, there was a clinically meaningful increase in walking speed of 0.04 m/s [19]. Although not statistically significant, both the mental and physical HRQoL domains improved with effect sizes (Cohen’s d) ranging from 0.41 to 0.52.

The values on all outcome measures at baseline and follow-up are summarized in Table 1.

### Adverse Events

There were few adverse events reported during this intervention. Two participants reported experiencing headaches during fasting periods, which resolved following an increase in water intake. One participant reported experiencing dizziness, which resolved after having a small snack.

## 4. Discussion

To our knowledge, this is the first study to examine the effect of a time restricted eating pattern in older adults (ages 65 and older). The primary findings of this pilot study were that a four-week TRF intervention was feasible and acceptable in overweight, sedentary older adults who previously reported fasting less than 12 h daily. Moreover, participants lost approximately 2.6 kg over this four-week intervention. Participants reported an eating pattern consistent with TRF on almost all days (6 out of 7) each week (mean adherence = 84%). During post-treatment interviews, most participants reported that this eating approach was acceptable and that they would be willing to maintain this type of eating pattern with some modifications, suggesting TRF is a practical strategy for many older adults.

The amount of weight loss participants achieved in this four-week intervention study of 2.6 kg is in line with previous findings on the effects TRF regimens have on changes in body weight in overweight middle-aged and younger individuals [20]. Moreover, most studies suggest TRF interventions produce significant reductions in body fat without significant loss of lean tissue in middle-age and younger individuals [20]. In line with this, Antoni et al. (2018) recently found that a 10-week TRF intervention significantly reduced body fat percentage, but did not significantly change body weight in healthy and overweight young and middle-age adults [21]. The participants in the Antoni et al. study were instructed to reduce their eating window by delaying the start date of their first meal by 1.5 h and moving the time of their last meal forward by 1.5 h. This approach resulted in an average reduction in their eating window of 4–5 h, which is in line with the reduction in the eating window reported by participants in our study.

Participants had a small but clinically meaningful increase in walking speed on the six-minute walking test, which measures the amount of distance the participant can complete on a standard walking course in six minutes without running or overexerting themselves [22]. This test is thought to be a better measure of exercise capacity than other shorter walking tests [19]. Since this is the first study to examine the effects of TRF on changes in physical function in older adults, these results should be considered preliminary and we cannot rule out the possibility of a practice effect on this measure [19,23]. Nevertheless, the improvement in walking speed is an encouraging sign as a large body of literature now indicates that walking speed is a strong predictor of major health outcomes and mortality in older adults [24,25,26], and that improvements in walking speed in older adults are associated with reduced risk of mortality [26]. Moreover, interventions that can enhance mobility and physical function in older adults may also improve overall quality of life [27].

Although not statistically significant, participants reported improvements of 5%–8% on the mental and physical health domains of the SF-12, a standard measure of HRQoL or perceived health status. The largest increase was shown on the physical health domain, suggesting the TRF intervention improved perceptions of physical function and ability to perform daily activities. To our knowledge, no studies have examined the effects of TRF interventions on HRQoL in overweight older adults, and few studies have examined the effects intermittent fasting regimens have on changes in HRQoL. Findings from a recent systematic review indicate “healthy” dietary patterns, marked by high consumption of vegetables, fruits and whole grains, legumes, seafood, and low consumption of sweetened foods, refined grains, and processed meats, are associated with better self-rated health and improvements in one or more domains of HRQoL [28]. The evidence is currently mixed; however, regarding the effects that calorie restriction interventions have on quality of life in overweight, older adults [29]. Thus, it is important to explore the potential effects that different types of eating patterns, such as intermittent fasting, have on changes in HRQoL in overweight older adults.

Recent findings by Kant and colleagues [30] indicate that most Americans consume food for greater than 12 h of the day, and thus fast for less than 12 h every day. In line with these findings, the participants in the present study reported fasting for less than 12 h per day at baseline. During the intervention period, participants reported fasting for approximately 15.8 h daily, which indicates they fasted for approximately 4 additional hours daily than at the time of enrollment. Some participants; however, were initially confused regarding what was acceptable to consume during the fasting and eating time periods. This suggests that weekly phone or in person contact is needed when delivering this intervention to older adults.

There were few adverse events reported during this intervention. Two participants reported experiencing headaches during fasting periods, which resolved following an increase in water intake. One participant reported experiencing dizziness, which resolved after having a small snack. Additional phone calls were made to these participants to monitor the events and ensure continued resolution. This study had some important limitations that should be noted, and thus findings of this study should be interpreted with much caution. First, the small sample size and short duration of the intervention limits the generalizability of the results and may have limited our ability to detect statistical significance. For this reason, we have reported the effect sizes (Cohen’s d) for all outcome measures since this statistic is independent of sample size. Second, there was no control or comparison condition so we cannot be sure that the observed changes in weight and quality of life were not due to some other factor. The amount of weight loss that we observed; however, is unlikely to have occurred without some form of dietary intervention based on previous trials in overweight individuals [20]. Another weakness of this trial is that we did not directly measure body composition and thus cannot make any conclusions about the type of weight that was lost. Finally, we did not assess dietary intake and thus are unable to determine the extent to which specific dietary patterns may have influenced the results of this study.

The present study also had a few strengths. First, adherence was carefully tracked throughout the study using an eating diary in which participants reported the time of their first and last meal each day. Participants received weekly phone calls to check on their adherence and assist them in problem-solving any challenges they were experiencing following the intervention. Second, during these calls, adverse events were assessed and recorded. We also conducted an exit interview to obtain the participant perspective on what challenges and what changes, if any, they would recommend to the intervention in future trials.

## 5. Conclusions

The findings of this pilot study suggest that TRF is an acceptable and feasible eating pattern for overweight, older adults to follow. Moreover, this eating pattern produced significant short-term weight loss and small but meaningful improvements in walking speed and HRQoL. The findings; however, should be interpreted very cautiously as the present study has a number of limitations. Future studies should examine the effects of this eating pattern using larger samples of older adults over longer time periods to better understand the effects of this promising intervention in this high-risk population.

## Figures and Tables

**Figure 1 nutrients-11-01500-f001:**
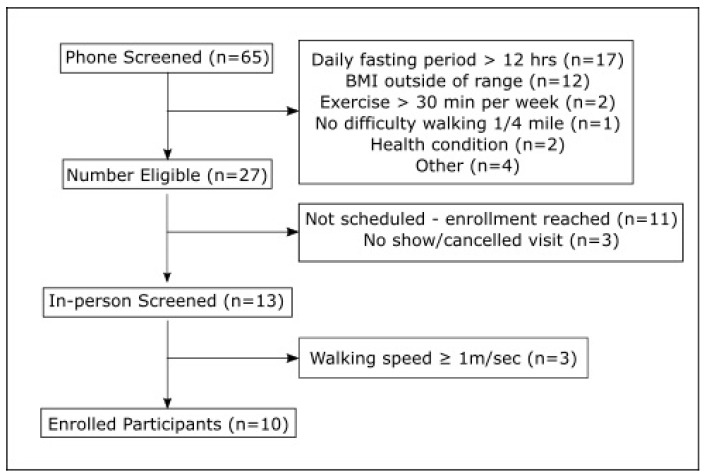
Participant recruitment flow.

**Table 1 nutrients-11-01500-t001:** Baseline and Follow-Up Values on All Study Measures.

Study Measures	Baseline *M (SD)*	Follow-up *M (SD)*	Cohen’s d (Effect Size)	*p*-Value
**Antropometric and Metabolic Measures**				
Body Weight	96.96 (16.2)	94.81 (16.9)	0.13	**0.009**
Body Mass Index (BMI)	34.1 (3.3)	33.2 (3.2)	0.29	**0.013**
Waist Circumference (cm)	109.43 (12.9)	109.23 (12.3)	0.02	0.602
Blood Glucose (mg/dL)	105.6 (28.2)	107.3 (29.4)	0.06	0.736
Systolic Blood Pressure (mmHg)	145.9 (15.6)	148.22 (24.2)	0.11	0.812
Diastolic Blood Pressure (mmHg)	78.1 (12.4)	78.89 (8.3)	0.07	0.877
**Physical Function**				
Six Min Walk (meters)	301.8 (91.0)	310.89 (111.2)	0.09	0.585
Six Min Walk (m/s)	0.88 (0.2)	0.92 (0.2)	0.21	0.877
Grip Strength (dominant hand)	22.3 (7.0)	24.0 (6.8)	0.13	0.270
**Health Related Quality of Life**				
SF-12 Physical Function (Summary Score)	13.6 (3.1)	14.9 (2.0)	0.52	0.185
SF-12 Mental Function (Summary Score)	22.0 (2.1)	22.8 (1.7)	0.41	0.285
SF-12 Total Score	35.6 (4.6)	37.7 (3.2)	0.54	0.170
**Fatigability**				
Pittsburgh Fatigue Scale (Mental)	13.8 (8.3)	14.7 (8.9)	0.10	0.787
Pittsburgh Fatigue Scale (Physical)	24.7 (6.8)	24.9 (8.2)	0.03	0.650
**Cogntive Function**				
MoCA	25.6 (3.4)	25.9 (3.1)	0.09	0.810

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
