# Peer review of "The Effects of Time Restricted Feeding on Overweight, Older Adults: A Pilot Study"

_nutrients, 2019, doi:10.3390/nu11071500_

Round 1
Reviewer 1 Report
The study was a short term TRF intervention conducted in overweight older adults with measures including changes in body weight , functional measurements with the main outcome being a significant body weight reduction.
Major limitations
The study although well written possesses a number of significant limitations including : lack of control group, lack of dietary intake data, and small sample size. Hence the findings should be interpreted with supreme caution. The lack of dietary intake data (despite food diaries being completed) is a significant limitation for a dietary intervention study. If the data are available, they must be analysed and presented.
Moreover the study introduction places significant emphasis on FFM/FM changes , however this was not assessed and so emphasis should instead be shifted towards the feasibility / practicality of TRF, and it’s effects on body weight and functional measures.
Another important point to mention is that the observed reduction in FFM with weight loss is now thought to be derived from the protein component of adipose NOT skeletal muscle. https://onlinelibrary.wiley.com/doi/10.1002/oby.22393
Minor points
35-36 make clear lifespan extension pertains to non human data
Discussion
219 how does the feedback regarding the TRf intervention compare to feedback obtained by either pilot studies eg Antoni et al 2018 https://www.cambridge.org/core/journals/journal-of-nutritional-science/article/pilot-feasibility-study-exploring-the-effects-of-a-moderate-timerestricted-feeding-intervention-on-energy-intake-adiposity-and-metabolic-physiology-in-freeliving-human-subjects/9C604826401917A6CAD9CD10B72FEA32
Authors can’t make reference to changes in FFM/FM from other studies as meaningless, as the present study didn’t measure fM/FFM and inferences can’t be made
Inclusion/exclusion
Waschronotype considered ?
95 ethics
Declaration of Helsinki statement
Stats
We’re normality tests first conducted to justify parametric tests ?
Which stats package used ?
Author Response
Dear Ms. Wang,
Thank you for your careful attention to our manuscript and for welcoming us to resubmit our manuscript to Nutrients. Below are our responses to the reviewers’ comments. The reviewers’ suggestions are in bold text, and our responses are in in black text. Within the manuscript, all changes have been noted. Thanks to the collaboration of the reviewers and your editorial board with our team, we believe our manuscript has been significantly improved. We hope that you in turn find the revised manuscript suitable for publication in Nutrients.
REVIEWER #1
The study was a short term TRF intervention conducted in overweight older adults with measures including changes in body weight, functional measurements with the main outcome being a significant body weight reduction.
Major limitations
The study, although well written, possesses a number of significant limitations including: lack of control group, lack of dietary intake data, and small sample size. Hence the findings should be interpreted with supreme caution. The lack of dietary intake data (despite food diaries being completed) is a significant limitation for a dietary intervention study. If the data are available, they must be analysed and presented.
Response: We thank the reviewer for these important points and have now listed lack of dietary intake data, in addition to the other significant limitations, in the Discussion of this manuscript.
Specifically, we now state: “Finally, we did not assess dietary intake and thus are unable to determine the extent to which specific dietary patterns may have influenced the results of this study.”
Moreover the study introduction places significant emphasis on FFM/FM changes, however this was not assessed and so emphasis should instead be shifted towards the feasibility / practicality of TRF, and it’s its effects on body weight and functional measures.
Response: We appreciate the reviewer’s comment here and accordingly have revised the text to shift the focus much more on the feasibility/practicality of TRF, rather than FFM/FM changes in the present study.
Another important point to mention is that the observed reduction in FFM with weight loss is now thought to be derived from the protein component of adipose NOT skeletal muscle. https://onlinelibrary.wiley.com/doi/10.1002/oby.22393
Response: We thank the reviewer for sharing this important recent perspective paper with us and have carefully reviewed the article. Based on the model described in this paper, we agree that the loss of FFM that has reportedly occurred with calorie restriction is likely due, at least to some degree, to the protein component of adipose tissue.
As recommended, we have shifted the focus of this paper away from the potential loss of FFM with calorie restriction vs. intermittent fasting in this manuscript. We will, however, but be sure to carefully evaluate the components of FFM loss in future studies.
Minor points
35-36 make clear lifespan extension pertains to non-human data
Response: We have inserted “non-human” in this sentence to clarify that these findings come from non-human data.
Discussion
219 how does the feedback regarding the TRF intervention compare to feedback obtained by either pilot studies eg Antoni et al 2018 https://www.cambridge.org/core/journals/journal-of-nutritional-science/article/pilot-feasibility-study-exploring-the-effects-of-a-moderate-timerestricted-feeding-intervention-on-energy-intake-adiposity-and-metabolic-physiology-in-freeliving-human-subjects/9C604826401917A6CAD9CD10B72FEA32
Response: This is an important question since the goal of the Antoni et al. study was to explore the feasibility of a time-restricted feeding (TRF) intervention in young and middle-age (pre-dominantly female) adults, across a broad BMI range (20 – 39.9 kg/m2) over a 10-week period. The participants in the Antoni et al. study were instructed to reduce their eating window by delaying the start date of their first meal by 1.5 hours and pushing back the time of their last meal by 1.5 hours. This approach resulted in an average reduction in their eating window of 4-5 hours, which is in line with the reduction in eating window in our study.
Our findings are in line with Antoni et al. which also found TRF was a feasible intervention and produced a significant amount of weight loss. Levels of adherence and retention to the TRF intervention were similar in both studies. Additionally, it appeared that approximately half of the study participants in both our study and Antoni et al., indicated that they would be willing to continue with this type of eating pattern.
A few noteworthy differences are that our study was the first to test the feasibility of TRF in older adults, and also included an objective assessment of physical function and self-reported quality of life, both of which were improved following the intervention. A noted weakness of our study is that we did not measure body composition, whereas Antoni et al (2018) study did assess body fat percentage using bioimpedance and found that participants significantly reduced their body fat percentage (but not body weight) after the 10-week TRF intervention.
We have now referenced the findings of this important study in the discussion section by stating
the following:
“Antoni et al. (2018) recently found that a 10-week TRF intervention significantly reduced body fat percentage, but not significantly change body weight, in healthy and overweight young and middle-aged adults.[1] The participants in the Antoni et al. study were instructed to reduce their eating window by delaying the start date of their first meal by 1.5 hours and moving the time of their last meal forward by 1.5 hours. This approach resulted in an average reduction in their eating window of 4-5 hours, which is in line with the reduction in eating window in our study.”
Authors can’t make reference to changes in FFM/FM from other studies as meaningless, as the present study didn’t measure fM/FFM and inferences can’t be made
Response: As recommended, we have now significantly decreased the emphases placed on changes in FFM/FM, specifically in regards to comparisons between calorie restriction vs. intermittent fasting, throughout the paper. We will, however, be sure to carefully evaluate the components of FFM loss in future TRF studies.
Inclusion/exclusion
Was chronotype considered?
Response: For this study, chronotype was not considered as part of the eligibility criteria.
95 ethics Declaration of Helsinki statement
Response: We have added a Declaration of Helsinki statement to the Ethics statement.
Stats
Were normality tests first conducted to justify parametric tests?
Which stats package used?
Response: We thank the reviewer for this important point. Normality tests were not previously conducted prior to conducting parametric tests. However, we have now conducted normality tests and explored normality plots, which indicated all variables were normally distributed. The stats package used was SPSS, version 25.

Reviewer 2 Report
Overall, the paper is intriguing and has good merit in progressing the current knowledge of intermitted fasting. I understand this was a pilot study to lay the foundation for additional research studies. However, several methodological issues need addressing. See below:
1) Adherence of 14 to 18 hours of fasting per day is almost not even fasting.
For example: If I eat breakfast at 8 AM, lunch at noon, and dinner at 5 PM (or even 6 PM). I will have stayed within the fasting window. So basically there is minimal change to an individuals time of caloric intake.
2) Why only count weeks 2-4 instead of the entire four weeks? Since the fasting periods were not stringent at all this should not have have been an issue.
3) When assessing weight loss through time-restricted feeding, body composition is essential. Especially since determining muscle loss is vital in older populations.
4) Lines 261-265 was previously stated in lines 207-210 and should be removed.
Author Response
REVIEWER #2
Overall, the paper is intriguing and has good merit in progressing the current knowledge of intermitted fasting. I understand this was a pilot study to lay the foundation for additional research studies. However, several methodological issues need addressing. See below:
1) Adherence of 14 to 18 hours of fasting per day is almost not even fasting.
For example: If I eat breakfast at 8 AM, lunch at noon, and dinner at 5 PM (or even 6 PM). I will have stayed within the fasting window. So basically there is minimal change to an individuals time of caloric intake.
Response: We appreciate the reviewer’s perspective regarding the fasting goal range for this study. An important point here is that the daily fasting target of 16-hours is in line with the target fasting range that has been used in previous time restricted fasting studies in young and middle age adults (Anton et al. 2017; Antoni et al., 2018). Recent population surveys indicate that the typical American eats for greater than 12 hours per day, and thus fasts for less than 12 hours per day (Kant 2018). Specifically, Kant and colleagues (2018) in her article entitled “Eating patterns of US adults: Meals, snacks, and time of eating” analyzed eating patterns of U.S. adults, using data from NHANES 2009-2014. Based on their analyses, the length of a typical ingestive period for US adults is 12.2 hours."[2]
In line with the findings of Kant (2018), the participants in the present study reported fasting for less than 12 hours every day at baseline. The daily fasting goal for this study was 16 hours. In line with this, participants reported fasting for approximately 15.8 hours per day during this four week intervention. This indicates that participants fasted for approximately 4 additional hours daily than they were fasting at the time of study enrollment.
A big reason for testing the effects of this type of modest but achievable time restricted eating schedule, however, is because of its potential to be a sustainable way of eating. In line with this, we provided a goal range of 14-18 hours to allow for some daily flexibility and to increase the feasibility of TRF in the target population of older adults.
2) Why only count weeks 2-4 instead of the entire four weeks? Since the fasting periods were not stringent at all this should not have been an issue.
Response: The reason we did not include the first week as part of the adherence calculation for this study is because we asked participants to gradually increase the amount of time fasting over the first week. As noted above, all participants reported fasting for less than 12 hours per day prior to enrolling in this study. Thus, they were instructed to gradually increase their daily fasting time during the first few days of this study and to be in the target fasting range of 14 – 18 hours starting at Day 4.
3) When assessing weight loss through time-restricted feeding, body composition is essential. Especially since determining muscle loss is vital in older populations.
Response: We fully agree that this was a limitation of the present study and have noted this in the discussion. We have also noted the purpose of the present study was to test the feasibility of a time restricted feeding eating pattern in an older population. We will plan to assess body composition in future studies evaluating time restricted feeding.
4) Lines 261-265 was previously stated in lines 207-210 and should be removed.
Response: We thank the reviewer for catching this redundancy and have now removed lines 261-265.

Reviewer 3 Report
While the need to assess the feasibility of TRF in overweight adults is reasonable, a simple 4-week study does not provide much information about the major challenge to nutritional programs for health improvement (i.e. long-term adherence). Additionally, the need to assess the safety of eating within 8 hours per day is arguably questionable (this is not a very extreme dietary intervention and is easily within the capacity of the human body with the exception of certain specific medical conditions). The small sample size, lack of body composition assessment, short duration, no control group or randomization, and limited variables examined substantially limit the impact of this study. Nonetheless, the present communication does provide a small amount of information that may help inform future TRF work.
The authors are particularly commended for pre-registering their study and accurately providing information in the manuscript (i.e. the trial registration and manuscript seem to be in agreement regarding the a priori purpose of the study, etc., which is not always the case unfortunately).
Specific comments are included below.
Abstract
It is stated that the preservation of fat-free mass is particularly important, but it does not appear that FFM was even assessed.
Intro
The second sentence of the first paragraph should cite research supporting its claims (i.e. evidence that overweight older adults are at an increased risk of functional decline).
Second paragraph: the research cited in the first sentence is over 20 years old even though the sentence begins with “to date…”. Have there been any changes in the last 20 years? If so, is there a more updated analysis that can be referenced? A reference this old does not address the literature “to date.”
Line 48: Intermittent fasting does not need to be capitalized.
Line 68: review articles summarize the findings of other researchers. I am not sure if it is appropriate to state that “we found that TRF interventions…produce significant reductions in body fat without significant loss of lean tissue…” unless you performed a meta-analytical analysis. The researchers who performed the original data collections found this, and the review article simply summarized the original data in a narrative form (unless meta-analysis was employed).
Lines 75 – 76: It is stated that the primary purpose of this study was to evaluate the safety and feasibility of the TRF protocol. Is there a rationale for why it would not be feasible or safe to eat within 8 hour per day? Don’t many individuals regularly eat within 8 hours per day (e.g. those who skip breakfast, those who don’t eat after an early dinner, etc.).
Methods
Beginning at line 144: The format of the outcomes section doesn’t seem to read like a traditional manuscript. Although and Editor may need to confirm whether this is necessary, I would recommend revising the format of this section to be more similar to traditional manuscript formats. As it currently exists, it seems like a list that was copied and pasted from some type of procedures document.
Line 146: which scale was used?
Line 148: was waist circumference taken in duplicate (or triplicate), or was it a single estimate? Did the same researcher perform all baseline and week 4 assessments?
Discussion
As noted by the authors, one of the most novel aspects of this intervention is the utilization of older adults. This is one of the main reasons the research does represent a contribution to the literature despite its limitations.
Line 224: There appears to be an in-text reference in a different format than the others (superscripted #16).
Line 228: Due to differences in the quantity and quality of ADF and TRF research, as well as different participants, procedures, etc., it is still quite speculative whether there is a meaningful difference in the composition of weight loss between ADF and TRF. I would recommend softening this statement.
Line 259: recommend rewording “what was ok” to be phrased more formally (e.g. “what was acceptable”).
The limitations section does correctly identify the major limitations of the trial.
Author Response
REVIEWER #3
While the need to assess the feasibility of TRF in overweight adults is reasonable, a simple 4-week study does not provide much information about the major challenge to nutritional programs for health improvement (i.e. long-term adherence). Additionally, the need to assess the safety of eating within 8 hours per day is arguably questionable (this is not a very extreme dietary intervention and is easily within the capacity of the human body with the exception of certain specific medical conditions). The small sample size, lack of body composition assessment, short duration, no control group or randomization, and limited variables examined substantially limit the impact of this study. Nonetheless, the present communication does provide a small amount of information that may help inform future TRF work.
The authors are particularly commended for pre-registering their study and accurately providing information in the manuscript (i.e. the trial registration and manuscript seem to be in agreement regarding the a priori purpose of the study, etc., which is not always the case unfortunately).
Response: We thank the reviewer for these comments.
Specific comments are included below.
Abstract
It is stated that the preservation of fat-free mass is particularly important, but it does not appear that FFM was even assessed.
Response: The reviewer is correct that we did not assess FFM in the present study as the goal was to test the feasibility of a time restricted eating pattern in an older adult population. For this reason, we have removed any reference to FFM in the abstract.
We fully appreciate that this was a limitation of the present study and have noted this in the discussion. As we noted in our response to Reviewer #2, we plan to assess body composition in future studies evaluating time restricted feeding.
Intro
The second sentence of the first paragraph should cite research supporting its claims (i.e. evidence that overweight older adults are at an increased risk of functional decline).
Response: We have now added the following citations to this sentence.
Anton SD, Karabetian C, Naugle K, Buford TW. Obesity and diabetes as accelerators of functional decline: can lifestyle interventions maintain functional status in high risk older adults? Exp Gerontol 2013;48(9):888-97.
Samper-Ternent R, Al SS. Obesity in Older Adults: Epidemiology and Implications for Disability and Disease. Rev Clin Gerontol 2012;22(1):10-34.
Second paragraph: the research cited in the first sentence is over 20 years old even though the sentence begins with “to date…”. Have there been any changes in the last 20 years? If so, is there a more updated analysis that can be referenced? A reference this old does not address the literature “to date.”
Response: As recommended, we have now replaced the previous reference with a more recent reference (shown below) after this sentence.
Richardson A, Austad SN, Ikeno Y, Unnikrishnan A, McCarter RJ. Significant life extension by ten percent dietary restriction. Ann N Y Acad Sci 2016;1363:11-7.
Line 48: Intermittent fasting does not need to be capitalized.
Response: We have corrected this in the manuscript.
Line 68: review articles summarize the findings of other researchers. I am not sure if it is appropriate to state that “we found that TRF interventions…produce significant reductions in body fat without significant loss of lean tissue…” unless you performed a meta-analytical analysis. The researchers who performed the original data collections found this, and the review article simply summarized the original data in a narrative form (unless meta-analysis was employed).
Response: We appreciate the reviewer’s point here and have modified the language accordingly to clarify the source of the original data collections.
Lines 75 – 76: It is stated that the primary purpose of this study was to evaluate the safety and feasibility of the TRF protocol. Is there a rationale for why it would not be feasible or safe to eat within 8 hour per day? Don’t many individuals regularly eat within 8 hours per day (e.g. those who skip breakfast, those who don’t eat after an early dinner, etc.).
Response: The primary purpose of this study was to evaluate the safety and feasibility of the TRF eating pattern in an older adult population (age > 65 years). Previous studies have shown this approach to be safe and feasible in younger and middle-aged adults. To our knowledge, however, no study has examined the safety and feasibility of TRF in adults 65 years and older.
Based on previous studies, we hypothesized that it would be safe and feasible for older adults, who are not practicing TRF, to begin practicing this type of protocol. The first step in evaluating this, however, is to conduct a feasibility study.
Methods
Beginning at line 144: The format of the outcomes section doesn’t seem to read like a traditional manuscript. Although and Editor may need to confirm whether this is necessary, I would recommend revising the format of this section to be more similar to traditional manuscript formats. As it currently exists, it seems like a list that was copied and pasted from some type of procedures document.
Response: We have revised the format to be more similar to traditional manuscript formats.
Line 146: which scale was used?
Response: The type of scale that was used was a Detecto scale. We have inserted this in the Methods now.
Line 148: was waist circumference taken in duplicate (or triplicate), or was it a single estimate? Did the same researcher perform all baseline and week 4 assessments?
Response: Waist circumference was measured twice at each visit, and a third time if the first two measurements were not within 12.5px of each other. There was more than one research coordinator on this study, and thus the same research coordinator did not perform all baseline and week 4 assessments.
Discussion
As noted by the authors, one of the most novel aspects of this intervention is the utilization of older adults. This is one of the main reasons the research does represent a contribution to the literature despite its limitations.
Response: We thank the reviewer for this comment and agree that one of the most important novel aspects of this study was the inclusion of older adults.
Line 224: There appears to be an in-text reference in a different format than the others (superscripted #16).
Response: We have now corrected this in-text citation.
Line 228: Due to differences in the quantity and quality of ADF and TRF research, as well as different participants, procedures, etc., it is still quite speculative whether there is a meaningful difference in the composition of weight loss between ADF and TRF. I would recommend softening this statement.
Response: As recommended, we have softened this statement.
Line 259: recommend rewording “what was ok” to be phrased more formally (e.g. “what was acceptable”).
Response: We have now revised the wording accordingly.
The limitations section does correctly identify the major limitations of the trial.
Response: We have now revised this section to more adequately describe the major limitations of the present trial (i.e., lack of control group, lack of dietary intake data, small sample size, and short-term intervention). We have further noted the findings should be interpreted with caution.
Reference List
[1] Antoni R, Robertson TM, Robertson D, Johnston JD. A pilot feasibility study exploring the effects of a moderate time-restricted feeding intervention on energy intake, adiposity, and metabolic physiology in free-living human subjects. Journal of Nutritional Science 2018;7(e22):1-6.
[2] Kant AK. Eating patterns of US adults: Meals, snacks, and time of eating. Physiol Behav 2018;193(Pt B):270-8.

Round 2
Reviewer 1 Report
Thank you for addressing the points raised, I have no further comments
Author Response
We appreciate the reviewer's time in reviewing our manuscript and are happy to hear that our responses addressed the previous critiques.
Reviewer 2 Report
The changes made to your manuscript improved the overall delivery and quality. I highly suggest revising your study design before continuing with your future projects.
Author Response
We appreciate the reviewer's time in reviewing our manuscript and are happy to hear that our responses addressed the previous critiques. We believe that the reviewer's comments and suggestions have improved the overall quality of the manuscript. As suggested, we will definitely revise the study design in future projects.